# An Immersible Microgripper for Pancreatic Islet and Organoid Research

**DOI:** 10.3390/bioengineering9020067

**Published:** 2022-02-09

**Authors:** Eike Früh, Sebastian Bütefisch, Benjamin Gursky, Dennis Brüning, Monika Leester-Schädel, Andreas Dietzel, Ingo Rustenbeck

**Affiliations:** 1Institute of Pharmacology, Toxicology and Clinical Pharmacy, Technische Universität Braunschweig, D38106 Braunschweig, Germany; e.frueh@tu-bs.de (E.F.); d.bruening@tu-bs.de (D.B.); 2PVZ-Center of Pharmaceutical Engineering of the Technische Universität Braunschweig, Technische Universität Braunschweig, D38106 Braunschweig, Germany; b.gursky@tu-bs.de (B.G.); m.leester@tu-bs.de (M.L.-S.); 3PTB–Physikalisch-Technische Bundesanstalt/Federal Institute of Metrology, D38116 Braunschweig, Germany; sebastian.buetefisch@ptb.de; 4Institute of Microtechnology, Technische Universität Braunschweig, D38124 Braunschweig, Germany

**Keywords:** cytosolic calcium, electrophysiology, pancreatic islet, microgripper, organoids, SU-8

## Abstract

To improve the predictive value of in vitro experimentation, the use of 3D cell culture models, or organoids, is becoming increasingly popular. However, the current equipment of life science laboratories has been developed to deal with cell monolayers or cell suspensions. To handle 3D cell aggregates and organoids in a well-controlled manner, without causing structural damage or disturbing the function of interest, new instrumentation is needed. In particular, the precise and stable positioning in a cell bath with flow rates sufficient to characterize the kinetic responses to physiological or pharmacological stimuli can be a demanding task. Here, we present data that demonstrate that microgrippers are well suited to this task. The current version is able to work in aqueous solutions and was shown to position isolated pancreatic islets and 3D aggregates of insulin-secreting MIN6-cells. A stable hold required a gripping force of less than 30 μN and did not affect the cellular integrity. It was maintained even with high flow rates of the bath perfusion, and it was precise enough to permit the simultaneous microfluorimetric measurements and membrane potential measurements of the single cells within the islet through the use of patch-clamp electrodes.

## 1. Introduction

2D cell culture systems have been the mainstay of biomedical research for at least the last forty years because they offer the availability of a variety of cells in large quantities under standardized conditions. While this model of cellular physiology is appropriate for exploring the basic regulation at the molecular-and-subcellular level, it is insufficient to explain the regulation taking place at the tissue-and-organ level, where cells of different functions interact [1]. With regard to drug development, the 3D cell culture increases the probability of correctly predicting the drug action at the organ, or the organismic, level [2]. However, the advantages of 3D cell culture models can only be brought to fruition when the instrumentation is available to handle them in a precise and gentle manner. 

Remote-controlled microgrippers have appeared as a solution to this task, in particular when analytical methods, such as live-cell imaging or patch clamping, are intended, where the 3D cell aggregates have to be held in a constant position during the experiment. A limited variety of grippers for microscopic biological objects have been developed [3,4,5]. The intended use is often decisive for the choice of the actuation principle. Electrostatic, thermal, electromagnetic, and piezoelectric actuations are the most commonly used principles [6]. However, electrically driven microgrippers are not easily compatible with work in aqueous media, as would be required for work with 3D cell aggregates. Moreover, electromagnetic fields could disturb the measurement of the very weak electrical signals (in the picoampere range) during the electrophysiological measurements. Thermomechanical actuation, such as the shape memory alloy design [7,8], or the thermal expansion design [9], has the inevitable disadvantage of heat dissipation, which may impair the physiological function under study. These considerations limit the range of actuation principles that are compatible with the intended bioanalytical procedures to pneumatic, hydraulic, or mechanic actuations. 

The MIN6-pseudoislet was chosen as the 3D-cell-culture system to be handled by the microgripper. MIN6-cells are immortalized insulin-secreting cells, which share many similarities with the pancreatic beta-cells, but have a much higher tendency to proliferate [10]. Under appropriate culture conditions, the MIN6-cells aggregate and form so-called “pseudoislets”, which have a higher secretory response to insulinotropic stimuli than the single cells [11]. For further studies, pancreatic islets were used. The pancreatic islet is a naturally occurring miniorgan of spheroidal shape once it has been isolated from the surrounding exocrine pancreatic tissue. Pancreatic islets contain multiple cell types, most notably the beta-cells that synthesize and release insulin, the main glucoregulatory hormone of the human body [12]. The research on pancreatic islets is of major importance to the development of more effective treatments for diabetes mellitus, a chronic metabolic disease that causes hardship and suffering on a worldwide scale [13].

## 2. Materials and Methods

### 2.1. Chemicals 

Collagenase P from Roche (Sigma-Aldrich, Taufkirchen, Germany) was used for the islet isolation. Diazoxide and tolbutamide were from Sigma-Aldrich, and Fluo 4 was obtained from Thermo-Fisher Scientific (Schwerte, Germany). The cell culture medium, RPMI 1640 (without glucose), was from Sigma-Aldrich, and the fetal bovine serum (FCS Gold ADD) was from Bio&Sell (Nürnberg-Feucht, Germany). The bovine serum albumin (BSA, fraction V) and all other reagents of analytical grade were from E. Merck (Darmstadt, Germany). 

### 2.2. Design of the Microgrippers 

The design of the microgripper is based on the microgripper toolbox developed at the Institute of Microtechnology, of the Technische Universität Braunschweig, Braunschweig, Germany, for technical and life science applications [3,4,7]. The microgripper consists entirely of SU-8, a UV-sensitive, high-contrast, epoxy-based negative-tone photoresist. Because of the lithographic exposure to UV radiation and the subsequent development and curing, the resin polymerizes. It is then fluid resistant and highly transparent. The total length of the gripper is 7.2 mm, and the length of the gripper jaws is about 700 µm. The actuator’s linear positioning movement is transmitted via an integrated gripper gear to the gripper jaws, in a way so that they move evenly and parallel to each other, which has a centering effect.

Initially, a pneumatic actuator was investigated to avoid interference with the electrophysiological measurements by the drive that opens and closes the gripper jaws. A diaphragm-based pneumatic actuator concept was developed, and its fabrication was investigated. This actuator concept can eliminate air leakage, which was considered to be a disadvantage of a previously developed bellows actuator [6], when the gripper is immersed in water. This actuator consists of a small plastic tube closed on one side by an elastic rubber diaphragm, while compressed air can be supplied through a tube on the other side. The outward bulging of the membrane results in a force acting on the application point of the gripper gear.

To measure the force exerted on the tissue, force-sensing microgrippers were developed and tested. The initial version, which employed a Vernier-style gauge as readout, has been described in detail in a preceding communication [4]. Consistent with the simulations during drafting, the movement of the force-indicating gauge proved to be linear over a sufficiently large range. Two variants were produced, with sensitivities of 5 µm per 1.5 mN or 0.5 mN, respectively. 

More sensitive versions were investigated for the present study, with both reduction in the spring stiffness and amplification in the deflection. The highly sensitive force sensor system has considerably longer spring bars to make it as soft as possible. In order to not extremely increase the overall dimensions of the gripper, the longer spring structures are no longer located at the front of the gripper, but at the side, next to the gearbox. To increase the deflection by about 20%, a pointer is located in the center of one of the springs. While the gripper jaw makes a translational movement relative to the base of the force measurement structure, the pointer makes a rotational movement. Because of the length of the pointer, its tip deflects further than the displacement of the gripper jaw itself. This results in a deflection in the range of a few micrometers per 10 µN. This sensitivity can be considered to be at the limits of a mechanically stable design. 

Finally, a Bowden-cable-based actuator principle was tested. This allows the force to be generated distant from the gripped sample, which minimizes possible electrical influences when using a piezo actuator, for example. The Bowden cable consists of a fine wire, which is guided through the interior of a commercially available cannula. The cannula is bonded to the base of the central SU-8 component, and the wire is bonded to the force application point of the gripper gear. Using a stepper motor with a stepping width of 5.625 degrees, and a reduction gear (ratio of 82.25 to 1) acting on a precision positioner, resulted in 5456 steps per turn of the micropositioner, which is equivalent to a movement of 500 µm. This back-and-forth movement was transduced by the microgripper gear into a lateral movement of the jaws at a ratio of 1 to 4. 

### 2.3. Fabrication of the Microgrippers 

The microgripper was manufactured by lithography in a clean room. For this process, one lithography mask is required. It starts with a silicon wafer with a 360-µm thickness as a carrier. For better adhesion, the wafer is sputter-coated with a 10-nm-thick chromium layer before a 2.1-µm-thick copper layer is sputter-deposited, which serves as a sacrificial layer to allow the final separation of the microgrippers from the carrier wafer. After the oxygen plasma activation, the first layer of the precursor SU-8-50 (Kayaku Advanced Materials, Inc., Westborough, MA, USA) is deposited by spin coating. The sample is covered and left for 20 min at room temperature, and then for about 1 h at 60 °C for planarization, after which the cover is removed. The temperature is slowly increased to 100 °C and the SU-8 coating is left to dry for 7 h. The spin coating and drying are then repeated, which results in a second SU-8 layer, which leads to a total SU-8 thickness of about 400 µm. The negative photoresist SU-8 is exposed to a dose of about 1500 mJ/cm^2^ of broadband UV light and is subjected to a postexposure bake. To reduce the residual stress, the sample is left at room temperature for at least 12 h before development. In the last step, the sacrificial copper layer is alkaline-etched for several hours until all the structures have detached from the carrier wafer.

### 2.4. Islet Isolation and Tissue Culture 

Pancreatic islets were isolated from the pancreases of NMRI mice of either sex (12–14 weeks old) by injecting a collagenase solution (1.4 U per mL of Krebs–Ringer medium) into the common bile duct, transferring the excised pancreas into a Falcon tube, and incubating for 9.5 min at 37 °C. After shaking by hand, the resulting suspension was centrifuged (300 G) and washed twice with ice-cold Krebs–Ringer medium. Thereafter, the islets were hand-picked under a stereomicroscope (for details see reference [14]). The composition of the HEPES-buffered Krebs–Ringer medium was (mM): NaCl 118.5; KCl 4.7; CaCl_2_ 2.5; KH_2_PO_4_ 1.2; MgSO_4_ 1.2; NaHCO_3_ 20; HEPES 10; and BSA at a 0.2% *w/v*. If not stated otherwise, the glucose concentration was 5 mM. Islets were used either freshly isolated, or after a culture period of 18 h. The medium for the islet culture was RPMI 1640, containing 5 mM glucose, 10% fetal bovine serum, and penicillin/streptomycin. The animal care was supervised by the regional authority (LAVES, Lower Saxony, Germany) and conformed to the current EU regulations.

Insulin-secreting MIN6-cells (kindly provided by Jun-Ichi Miyazaki, [10]) were seeded on glass cover slips and cultured in DMEM medium (25 mM glucose), supplemented with 6 mM L-glutamine, 10% fetal bovine serum, and penicillin/streptomycin. The MIN6-pseudoislets [11] were generated by culturing MIN6-cells on Petri dishes for suspension in the cell culture. A total of 8–10 days were usually needed to obtain pseudoislets of sufficient size. The culturing of the NMRI islets and pseudoislets took place in a humidified atmosphere of 95% air and 5% CO_2_ at 37 °C.

### 2.5. Integrity of Freshly Isolated and Cultured Islets 

The relation of live and dead cells in the islets was assessed by the live/dead assay (PromoCell, Heidelberg, Germany). The membrane-permeable calcein/AM ester is cleaved in the cytosol of living cells, yielding a green fluorescence. Ethidium homodimer III is excluded from a living cell with an intact plasma membrane, but it can reach the nuclei of dead cells. Binding to the nuclear DNA intensifies the red fluorescence about fortyfold. After loading with the indicators for 40 min, the islets were placed on Petri dishes with glass bottoms (Ibidi GmbH, Gräfelfing, Germany) and were placed on the stage of an inverted Nikon Ti2-E microscope fitted with a Yokogawa CSU W1 spinning disk unit (Yokogawa Electric, Musashino, Japan). The green fluorescence of the calcein was excited at 491 nm, and the red fluorescence of the ethidium homodimer was excited at 561 nm, and it was collected by a Nikon CFI Plan Apochromat Lambda S40 XC Sil objective (40 × magnification, 1.25 N.A.). The images were acquired by a sCMOS camera (Prime BSI, Teledyne Photometrics, Tucson, AZ, USA) under the control of VisiView Premier software (Visitron Systems, Munich, Germany). 

### 2.6. Microfluorimetric Measurements 

The NMRI islets were loaded at 32 °C with Fluo4/AM at a concentration of 1 µM for 40 min. Thereafter, 5 islets were placed in the center of a Petri dish with a glass bottom (Ibidi GmbH, Gräfeling, Germany) on the stage of an Axiovert 135 microscope (Zeiss, Jena, Germany). The HEPES-buffered Krebs–Ringer medium was added via a Teflon tube and removed by a filament-containing cannula on the opposite site of the dish, and both the addition and removal were controlled by the same peristaltic pump. One islet was held by the microgripper and perifused with medium, which was continuously gassed with a mixture of 95% O_2_ and 5% CO_2_. The fluorescence was excited by a Xenon arc lamp at 490 nm, and the emission (>510 nm) was collected by a Zeiss LD AchroPlan with a 40× magnification objective, and was recorded by an UI-3060CP-M-GL R2 camera (IDS, Obersulm, Germany).

### 2.7. Electrophysiological Measurements 

The plasma membrane potential of a single cell within a microgripper-held islet or pseudoislet was measured using the patch-clamp technique. Pipettes were pulled from borosilicate glass (2 mm o.d., 1.4 mm i.d., Hilgenberg, Germany) by a two-stage vertical puller (HEKA-Electronics, Lambrecht, Germany) and had resistances between 3 and 6 MΩ when filled with solution. The measurements were performed using an EPC 7 patch-clamp amplifier (HEKA-Electronics), Labview 2015 software, and a PCI 7833 data acquisition board (National Instruments Germany, München, Germany). The data were stored on a hard disk and analyzed offline using GraphPad Prism5 software (GraphPad, LaJolla, CA, USA). A slow bath perfusion system was used, and all experiments were performed at room temperature (20–22 °C). For the measurements of the membrane potential in the perforated patch mode [15,16], the pipette solution contained (mM): 10 KCl; 10 NaCl; 70 K_2_SO_4_; 7 MgCl_2_; and 5 HEPES, with a pH of 7.15, plus 125 µg/mL of the pore-forming agent, nystatine (2.5% DMSO final concentration). The extracellular bath solution contained (mM): 140 NaCl; 5.6 KCl; 1.2 MgCl_2_; 2.6 CaCl_2_; 10 HEPES; and 1 glucose, with a pH of 7.4.

## 3. Results

The initially tested actuation principle was a closed pneumatic system because the constant generation of the air bubbles of the original open pneumatic design interfered with the microscopic observation. Increasing the air pressure within the system protruded the elastic membrane at the tip of a conical polyethylene tube by ca. 0.2 mm, which set the jaws of the microgripper into motion (Figure 1). However, functional testing revealed that the relation between the increase in the air pressure and the protrusion of the membrane was very steep, and resulted in a virtual all-or-nothing response characteristic (Appendix A). Gripping a normal mouse islet always led to severe deformation, which, in most cases, was irreversible. 

This experience demonstrated the necessity of obtaining a clearer view of the force needed to clamp an islet without causing its deformation or damage. To this end, a particularly sensitive force-indicating gripper was devised (Figure 2). The microscopically visible movement of the pointer relative to a fixed rod started at about 30 µN. However, even with this extremely sensitive design, the indicator rods only began to move when the gripping jaws exerted an irreversible deformation on a clamped islet (Figure 2). Since the gripping force necessary to move islets or pseudoislets around in the perifusion medium was always below the lowest level of determination, the conclusion was that the generation of a stable, but not detrimental, contact required the finely tunable movement of the jaws under visual control.

To achieve this characteristic, a mechanical actuation system was devised, where the jaws of the microgripper were set into motion by a stepper motor acting on a rotating fine positioner, which, in turn, acted on a Bowden-cable-like transmission system (Figure 3). With this actuation system, the movement of the jaws was smooth and the distance between the jaws could be precisely adjusted between 400 µm and a complete closure (Figure 3). The (theoretical) spatial resolution was a movement of the jaws of about 0.37 µm per step of the motor. This movement of the jaws enabled a deformation-free contact between the islet and the microgripper, which was sufficiently stable to permit back-and-forth movements within the Krebs–Ringer bath solution. 

First, the microgripper was lowered to a position just above the bottom of the 35-mm Petri dish on the microscope stage. The islets were pipetted in the middle of the Petri dish, and the microgripper was moved towards the islets with the jaws in the open position. The movement of the gripper did not make the islets flow away. When an islet was located between the gripper jaws, they were closed under visual control until a contact seemed likely, but no deformation had yet occurred. The stability of contact was tested by moving the gripper back and forth. In case the islet was not directly following the movement of the gripper, a slight further closing of the jaws was usually sufficient to establish the contact. 

The robustness of the deformation-free contact was tested by increasing the exchange of the bath solution. The normal flow rate was 1 mL/min, nominally equivalent to one complete exchange of the bath volume. Flow rates of up to 5 mL/min were tolerated without a visible change in the position of the pseudoislet islet held by the gripper, whereas pseudoislets that had grown attached to the bottom of the Petri dish were washed away at flow rates above 2 mL/min (Appendix A). Primary islets had a stronger adhesion to the Petri dish but, starting at flow rates of 4 mL/min, they too were washed away. It is likely that the slightly sticky surface of the islets contributes to the stability of the contact. Of note, this property is more prominent with freshly isolated islets. To ascertain that the contact between the gripper jaws and the islet was innocuous for the islet structure, live/dead assays were performed after holding the islets for 10 min in a perfused cell bath. A deformation-free hold, and even a slightly deforming hold, were without consequences as compared with the control islets (Figure 4). Only when the islet diameter was diminished by 30 to 50%, which is usually accompanied by an irreversible change in shape, did a clear increase in the necrotic cells in the superficial layer occur (Figure 4).

The visual control of the finely resolved gripping movement also enabled the gripping of MIN6-pseudoislets, which consist only of insulin-secreting MIN6-cells. In contrast to normal islets, they do not contain other endocrine cell types, vasculatures, or fibroblasts producing collagen. Therefore, the pseudoislets are mechanically much less stable than the primary islets. Placing a patch pipette (tip diameter ca. 1 µm) on one of the outer cells (Figure 5) and applying gentle suction led to the buildup of a Giga-Ohm seal as a prerequisite for performing the measurements of the membrane currents and potentials. The seal is easily disrupted by small movements of the cell surface relative to the tip of the pipette. However, the pseudoislet remained in a stable position during the bath perfusion at 2 mL/min, which enabled the registration of the membrane potential in the perforated-patch mode. The depolarizing effect of a high extracellular potassium concentration, and the generation of action-potential spiking by the addition of the K_ATP_ channel blocker, tolbutamide, could be followed for ca. 25 min (Figure 5).

It was also possible to measure the fluorescence of an islet held by the gripper jaws. Specifically, the depolarization-induced increase in the cytosolic Ca^2+^ concentration could be shown by using the fluorescence indicator, Fluo-4, which had been loaded into the islets prior to the experiment (Figure 6). Of particular interest is the possibility of combining electrophysiological measurements with live-cell imaging. Measuring the membrane potential of a peripheral islet cell together with the Fluo-4 signal of the entire islet showed the close correspondence of the action-potential spiking (which is caused by Ca^2+^ channels in beta-cells) and the increase in the cytosolic Ca^2+^ concentration (Figure 6). Furthermore, the connexon-mediated synchronization of the beta-cells within the islet became visible since all of the islet cells contributed to the fluorescence signal.

## 4. Discussion

The present design of a mechanically actuated microgripper permits the positioning of pancreatic islets, and other less stable multicellular aggregates, within cell culture media or other bath solutions. The position remains stable when the bath media are exchanged with reasonably high flow rates, thus permitting kinetic measurements. The actuator principle does not interfere with the sensitive measurements of the membrane currents and potentials, and the shape of the gripper jaws is compatible with live-cell imaging. 

The originally pursued pneumatic actuation was chosen to avoid the pick-up of electrical noise during the electrophysiological measurements [4]. Since the force transduction by a pneumatically vaulted membrane was not compatible with the finely tuned movement of the gripper jaws, the use of a stepper motor was reconsidered. While the use of stepper motors disturbed the electrophysiological measurements, it was the motor driver, not the stepper motor itself, which produced the electrical interference. Separating the components and placing the driver under the vibration isolation table provided sufficient shielding to enable the measurements of the membrane potential. 

Pancreatic islets and MIN6-pseudoislets are particularly suited to illustrating the advantages of measuring the reactions of 3D cellular aggregates, as compared with single cells. The different cell types of the pancreatic islet interact, and the secretion pattern of the alpha cells is subject to paracrine inhibition by beta- and delta-cells [12,16]. Even the comparatively homogeneous MIN6 pseudoislet cells illustrate this point since MIN6-cells within pseudoislets respond more strongly to metabolic stimulation than single MIN6- cells [11,17]. Similarly, the ability of tolbutamide to induce action potentials in the presence of 40 mM of KCl (see Figure 5) differs from the observations made under the same conditions in single beta-cells [18]. 

The microgripper permits the use of freshly isolated, as well as cultured, islets. With regard to the electrophysiological measurements, this is less trivial than it may appear since the conventional measurement of single islet cells requires a period of cell culture after the dissociation of the islet into single cells [19]. In the comparatively few studies on single cells within islets, collagenase digestion was often followed by a period of cell culture ([20], see also [21]). Since there is evidence that even short-term culture affects the response of the islets to nutrients such as glucose [22], investigations of freshly isolated islets may gain in importance.

If measurements of islet cells that have not spent time in a cell culture are intended, the slicing of the agarose-hardened pancreas and the identification of the slices containing islets is a viable option [23,24]. While this preparation offers the potential advantage that the innervation is better preserved, it requires more specialized equipment and is less versatile than the microgripper, which cannot only be used for islet research, but also for research on virtually any organoid of spherical shape. This is a relevant feature for the emerging field of islet spheroid reaggregation, either from primary human cells, or from stem cells [25,26,27].

In conclusion, the microgripper fulfills the requirements for a tool to handle 3D cellular aggregates or organoids during prolonged cell-physiological experimentation. It is compatible with the measuring stands that were designed for the work with single cells or the 2D monolayers of cells. In its current version, the microgripper is a tool for low-throughput techniques, but it increases the depth of information that can be gained from a complex multicellular aggregate. Similar to the conventional patch-clamp technique, its use requires manual skills and training. However, these requirements may become less critical with the development of automated object recognition, which may not only steer the microgripper, but also the measuring pipette. 

## Figures and Tables

**Figure 1 bioengineering-09-00067-f001:**
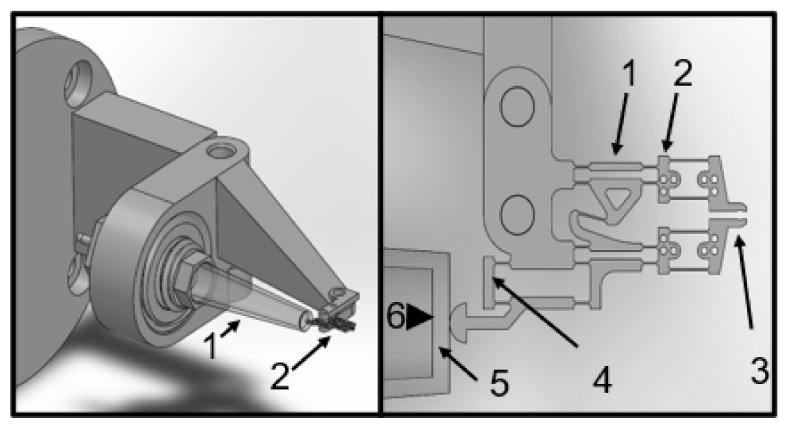
Design of a closed system for the pneumatic actuation of the gripper jaws. Increasing the air pressure protrudes the membrane at the tip of the tube whereby the jaws are set in motion. The left panel shows a 3D view of the entire setup. “1” denotes the membrane-sealed tube, and “2” denotes the microgripper. The right panel shows a detailed view of the gripper structure. The numbers denote the following: 1 = gear structure for generating a parallel movement of the gripper jaws; 2 = force measurement structures with optical markers; 3 = jaws; 4 = gear structure for the transmission of the membrane protrusion; 5 = actuator membrane (nonprotruding); and 6 = air pressure.

**Figure 2 bioengineering-09-00067-f002:**
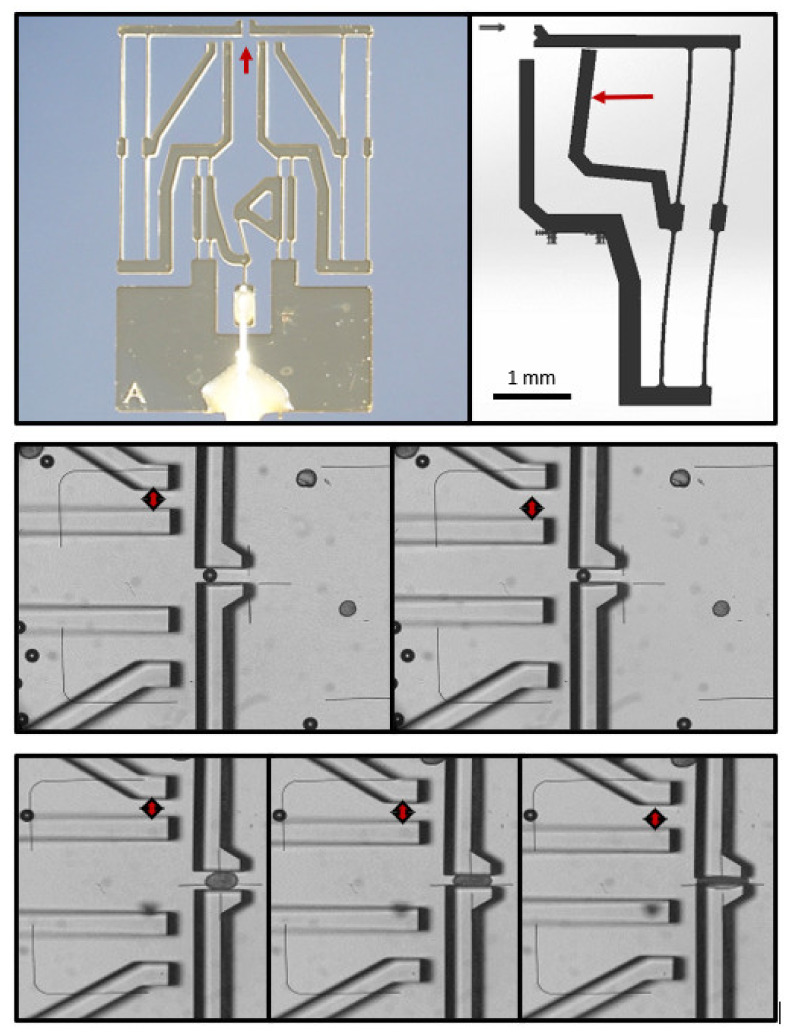
Measurement of the gripping force. (**Upper row**): overview of the entire microgripper with integrated force sensing (left), and schematic view of the principle of function (right). The arrow in the left panel points to the gap between the gripping jaws, and the one in the right panel indicates the deflected pointer. (**Middle row**): gripping of a noncompressible polyethylene microbead with minimal force (left), and with near maximal force (right). The microbead has a diameter of 100 µm, and the width of the gap increases from ca. 150 µm to ca. 200 µm, as indicated by the double-headed arrow. (**Lower row**): gripping with minimal force (left), reversible deformation by gripping (middle), and irreversible deformation by gripping (right). As indicated by the carets, only the force that irreversibly deforms the islet is sufficient to move the pointer by ca. 10%, which is equivalent to ca. 15 µm. This force can be estimated to correspond to ca. 30 µN.

**Figure 3 bioengineering-09-00067-f003:**
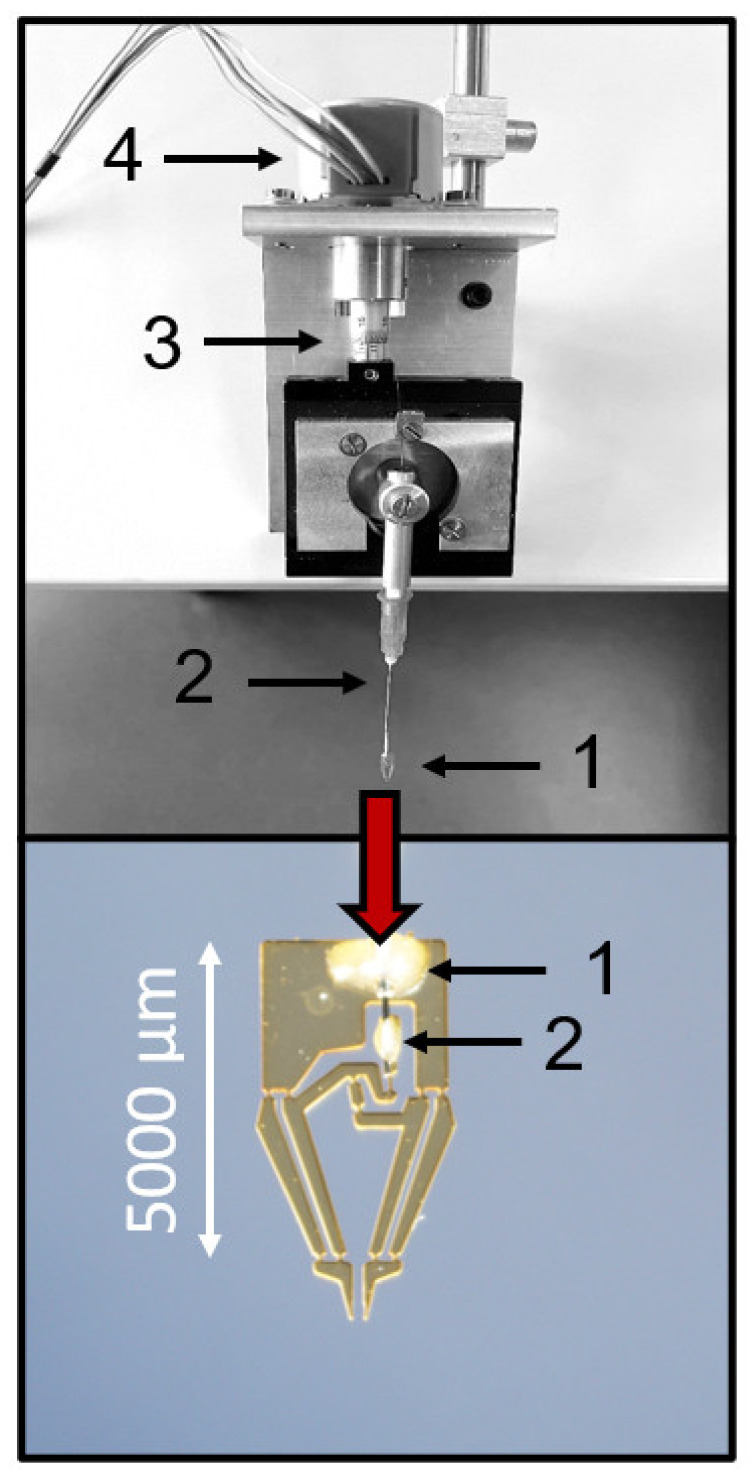
Mechanical actuation with force transduction via Bowden cable. The upper image shows the entire setup. The numbers denote the following: 1 = microgripper; 2 = cannula; 3 = micrometer screw; 4 = stepper motor. The lower image shows the microgripper in close-up view. “1” denotes the site where the cannula is bonded to the microgripper base, and “2” denotes the site where the wire guided through the cannula is bonded to the force application point of the gripper gear.

**Figure 4 bioengineering-09-00067-f004:**
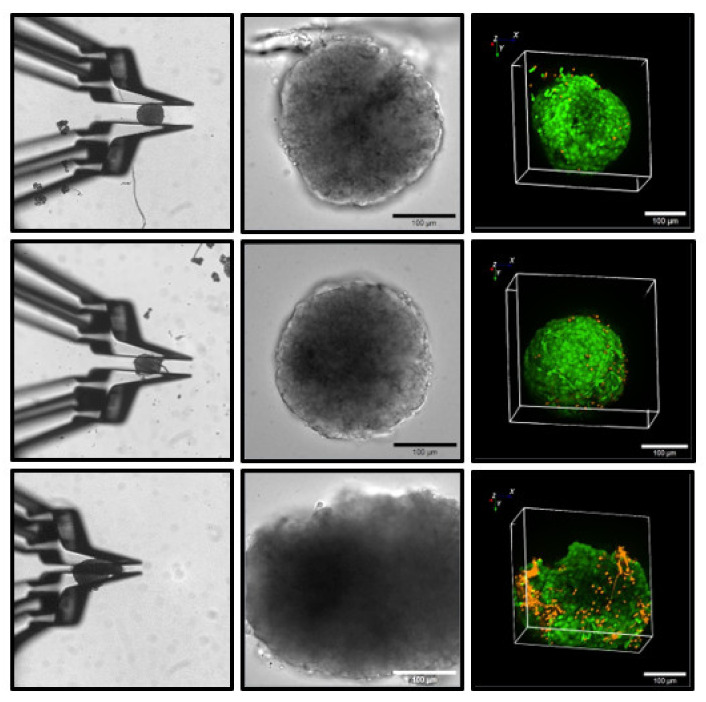
Absence of cellular damage by gripping with minimal force (**upper row**), and with reversibly deforming force (**middle row**), and widespread occurrence of necrosis after irreversible deformation (**lower row**). The left column of images shows the aspect of the islets during gripping, and the middle column shows the aspect of the islets after gripping, both in transmitted light. The right column shows the 3D aspect of the respective islets as registered by confocal laser scanning fluorescence. Intact cells are marked by green fluorescence, and dead cells are marked by red fluorescence. Gripping without visible deformation remains without cellular damage in the islets. The force exerted in the experiment shown in the lower row can be estimated to be ca. 30 µN (compared with Figure 2).

**Figure 5 bioengineering-09-00067-f005:**
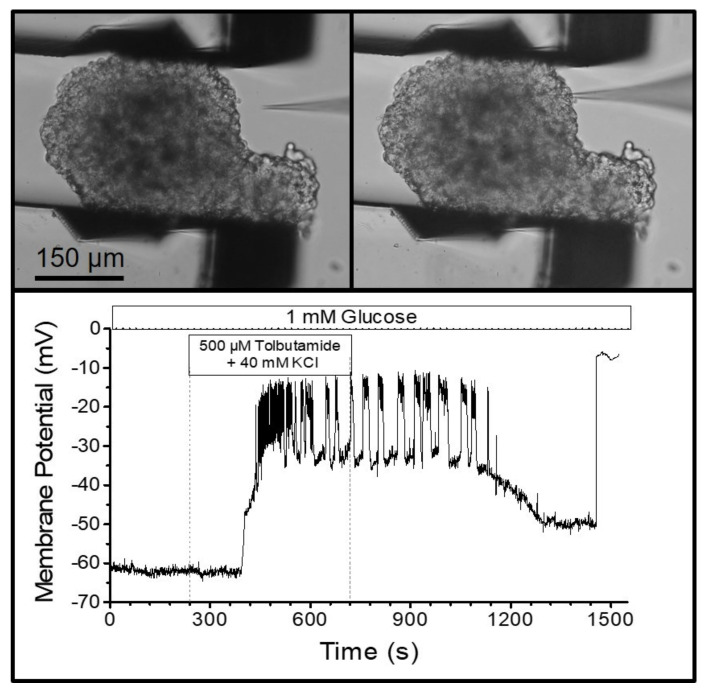
Measurement of the plasma membrane potential of an MIN6 insulin-secreting cell within a pseudoislet. The (**upper graph**) shows the irregularly formed pseudoislet with the tip of the patch pipette approaching (left), and the tip attached to a single cell after seal formation (right). The (**lower graph**) shows the resulting registration. The pseudoislet was continuously perifused with a medium containing 1 mM of glucose. From 240 to 720 s, the medium additionally contained 500 µM of tolbutamide, and the potassium concentration was increased to 40 mM. Note the occurrence of action-potential spiking, which stops upon washout.

**Figure 6 bioengineering-09-00067-f006:**
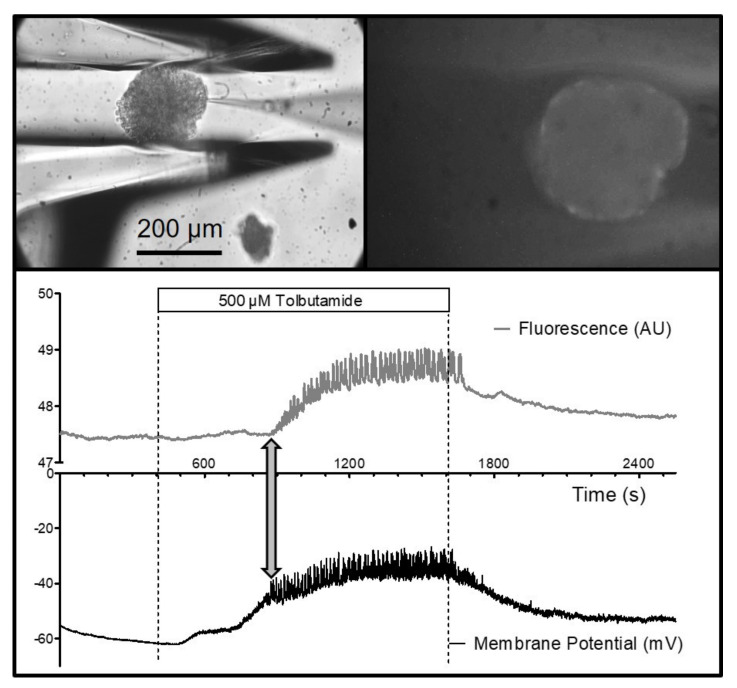
Measurement of the cytosolic Ca^2+^ concentration in a perifused islet concurrently with the measurement of the plasma membrane potential of a single islet cell. The (**upper graph**) shows the islet with the pipette tip attached to a single cell after seal formation (left), and the same islet emitting the fluorescence of the Ca^2+^ indicator, Fluo 4 (right). The (**lower graph**) shows the resulting registrations. The islet was continuously perifused with a medium containing 3 mM of glucose. From 400 to 1600 s, the medium additionally contained 500 µM of tolbutamide. Note the occurrence of action-potential spiking in the lower registration, which coincided with the increase in the cytosolic Ca^2+^ concentration (double-headed arrow). Upon washout, the spiking quickly ceased, as did the oscillatory pattern of the cytosolic Ca^2+^ concentration.

## Data Availability

The datasets generated and/or analyzed during the current study are not publicly available; however, they are available from the corresponding authors upon reasonable request.

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
