# Peer review of "An Immersible Microgripper for Pancreatic Islet and Organoid Research"

_bioengineering, 2022, doi:10.3390/bioengineering9020067_

Round 1

Reviewer 1 Report

The authors designed a microgripper with a stable hold to handle 3D cells/organoids. Interestingly, it could be used to detect the membrane potential and fluorescence simultaneously and achieve a great agreement. Overall, the manuscript fits the scope of the journal Bioengineering well. The research design is reasonable and the data is solid for supporting the authors’ conclusions. While minor revisions still should be done before publication. Below are the comments for your consideration.

  1. Line 172. Why the NMRI islets were loaded at 32 °C instead of 37 °C. Is the specific temperature for this islet?
  2. Line 189. Normally, the cell is sensitive to temperature, when the experiments were performed at 20-22 °C, I am wondering how long it could be kept the experimental statues. Is it possible to add a temperature controller?
  3. Page 217. Is the design for the control of the force 30 uN. Could it be smaller? As shown in Figure 4, can the author recognize the force for the three situations? Can the machine be used for the Young Modulus (stiffness) testing for the cells? And is there any automobile setup or button that can control the force and avoid the destruction of the cells?
  4. Page 289 and 304. Scale bar should be added in Figures 5 and 6.
  5. Page 301. Interestingly, the membrane potential is consistent with the fluorescence reaction. I am wondering is there any possibility for the quantitative relationship between the membrane potential and the fluorescence? For example, when you reduce the concentration of tolbutamide, will the machine detect the reducing membrane potential and the fluorescence as well? If two parameters could be associated in a quantitative way, that will expand the applications of the equipment.
  6. What’s part costs most of the equipment? Except for microgrippers, can all other parts be purchased in the market?
  7. Some typos should be corrected, such as page 162: “40fold”.
  8. The supplementary Video 1 cannot be played. May double-check it.

Reviewer 2 Report

Fruh et al have produced an excellent new piece of equipment to aid with islet interrogation which has clear application with other 3D cell cultures/organoids.  I commend the team for their excellent technical achievements.  I have only the following minor suggestions for improvment:

The abstract is well written and reflects well on the paper but should reference the imaging achievement and also concentrate a bit more description on the gripper itself.

General comment to improve the English throughout – eg “for the last decades” in the first line is inelegant

The intro might further expand on why pancreatic islets are a particularly important example of this.  For example the early work by Brissova who showed that insulin secretion from beta cells is much improved in 3D culture and that there is increasing appreciation in the field of the importance of studying the islet as a 3D functional unit:  Frontiers | Cell Heterogeneity and Paracrine Interactions in Human Islet Function: A Perspective Focused in β-Cell Regeneration Strategies | Endocrinology (frontiersin.org)

Section 2.2 – is IMT Institute for Microtechnology?

Whilst the mechanics of the microgripper were nicely explained could the authors elaborate a little on the user interface please – how does the operator control and visualise and know how much pressure is being applied?

The methodology for islet isolation is unclear (after collagenase injection were separation steps undertaken?  Reference 14 here is inadequate)

Section 2.6 – please elaborate on how the islets, once held in the microgripper, were perifused.  It’s not clear from this description as they start off just sitting in a petri dish.

RESULTS – please keep the language concise and scientific eg “However, it turned out that the relation..” and “virtual squashing” are inelegant

I would like a little detail as to how the islets were approached and picked up in the first place – do they need to be pipetted into eh region of the jaws?

Figure 5 is a fantastic result well done.  Was this easily re-producible?  How many times did you have to try (once you are competenet with the technology) and is it possible to re etst EP in the same islet in different conditions?

In Figure 6 it’s important to clarify that the EP was single cell but the fluorescence readout was islet wide (ie you were not able to achieve single cell imaging resolution).

It’s not clear why you chose tolbutamide and not a higher concentration of glucose to image your responses.
